# The Immunomodulatory Role of Hypoxic Tumor-Derived Extracellular Vesicles

**DOI:** 10.3390/cancers14164001

**Published:** 2022-08-18

**Authors:** Joel E. J. Beaumont, Nicky A. Beelen, Lotte Wieten, Kasper M. A. Rouschop

**Affiliations:** 1Department of Radiotherapy, GROW—School for Oncology and Reproduction, Maastricht University Medical Center+, 6229 HX Maastricht, The Netherlands; 2Department of Internal Medicine, GROW—School for Oncology and Reproduction, Maastricht University Medical Center+, 6229 HX Maastricht, The Netherlands; 3Department of Transplantation Immunology, GROW—School for Oncology and Reproduction, Maastricht University Medical Center+, 6229 HX Maastricht, The Netherlands

**Keywords:** hypoxia, cancer, immunosuppression, extracellular vesicles, immunotherapy, tumor microenvironment

## Abstract

**Simple Summary:**

Hypoxia, a characteristic of many cancer types, can suppress the antitumor effector functions of the adaptive and innate immune system. Tumor-cell-derived extracellular vesicles, which function as a mechanism of communication between tumor cells and immune cells, are also affected by hypoxia, and may drive immunosuppression. The aim of this review is to summarize the current knowledge on hypoxic cancer-cell-derived extracellular vesicles in immunosuppression, and to provide an overview of enriched factors (i.e., miRNA and proteins) in hypoxic tumor-derived EVs and their role in immunomodulation. This complete overview may indicate relevant directions for future research into the role of hypoxia in immunosuppression during cancer.

**Abstract:**

Tumor-associated immune cells frequently display tumor-supportive phenotypes. These phenotypes, induced by the tumor microenvironment (TME), are described for both the adaptive and the innate arms of the immune system. Furthermore, they occur at all stages of immune cell development, up to effector function. One major factor that contributes to the immunosuppressive nature of the TME is hypoxia. In addition to directly inhibiting immune cell function, hypoxia affects intercellular crosstalk between tumor cells and immune cells. Extracellular vesicles (EVs) play an important role in this intercellular crosstalk, and changes in both the number and content of hypoxic cancer-cell-derived EVs are linked to the transfer of hypoxia tolerance. Here, we review the current knowledge about the role of these hypoxic cancer-cell-derived EVs in immunosuppression. In addition, we provide an overview of hypoxia-induced factors (i.e., miRNA and proteins) in tumor-derived EVs, and their role in immunomodulation.

## 1. Introduction

Cancer is a devastating disease caused by the uncontrolled proliferation of cells, and is the second leading cause of death globally. According to the World Health Organization, approximately 1 in 6 deaths are due to cancer, with an estimated 10 million deaths worldwide in 2020 [1]. Furthermore, the incidence and mortality rates are expected to steadily increase over the coming years. During tumor development, cancer cells acquire 14 biological properties, known as the hallmarks of cancer which, among other traits, allow them to proliferate uncontrollably, spread to other organs, prevent cell death, adapt to the extreme changes in the TME, and evade the immune system [2].

In addition to providing protection against infections, the immune system is involved in the clearance of debris, initiation of tissue repair, and preventing tumor growth through the elimination of neoplastic cells. As such, transformed cells are selected for their capability to escape this constant immune surveillance, resulting in tumor development [3]. Cancer cells can produce various immunosuppressive factors and recruit regulatory immune cells to induce immunotolerance [4,5]. Furthermore, they can acquire characteristics to corrupt the immune system into providing a pro-tumorigenic environment, stimulating angiogenesis and metastasis [6].

Hypoxia, caused by the high use of oxygen during the energy-demanding continuous growth of the tumor and defective vasculature, is a common trait of solid tumors. It is associated with therapy resistance, a more malignant phenotype, and poor survival [7,8]. Increased tumor aggressiveness and therapy resistance are mediated by hypoxia-induced protection mechanisms such as the unfolded protein response and autophagy [7,8,9]. Additionally, hypoxia impairs the immune compartment of the TME and reduces antitumor immune responses [10]. Furthermore, hypoxia is a major factor that contributes to the immunosuppressive phenotype of cancer cells [10]. Hypoxic tumor cells release a repertoire of immunoregulatory molecules to suppress antitumor immune reactions and orchestrate an immune response beneficial to tumor progression [10]. In addition to the classical means of communication between tumor cells and the immune system, mediated by the production of soluble factors, extracellular vesicles (EVs) have gained considerable attention as mediators of this intercellular crosstalk [11].

EVs are nano-sized membrane vesicles that are produced by most cell types. They contain a complex molecular cargo composed of lipids, proteins, and nucleic acids that are transferred to distant cells for intercellular communication [11]. In recent years, the interest of the scientific community in EVs has increased tremendously, owing to both their role in physiological intercellular crosstalk and their important roles in various pathological processes, including neurodegenerative- and autoimmune diseases and cancer [12,13,14,15]. This makes EVs appealing targets for various applications, including as diagnostic and prognostic biomarkers, drug delivery vehicles, and novel therapeutic targets.

Depending on their biogenesis pathway, EVs can be subdivided into exosomes—produced by inward budding of the early endosome and subsequent release by fusion of the multivesicular endosome with the plasma membrane—and microvesicles (MVs), produced by direct budding of the plasma membrane [11]. Although these two types of biogenesis occur at different subcellular locations, they share various intracellular mechanisms and sorting machineries [16]. The overlap in size and cellular machineries complicates the distinction between EV subtypes, and specific markers for each subtype after secretion are absent [16]. Therefore, the term EVs will be used throughout this review to refer to both MVs and exosomes collectively.

EVs can affect target cells via at least two identified methods: The first is mediated via ligand–receptor interaction, where ligands expressed on the surface of EVs bind to the respective receptor on target cells. This type of intercellular communication does not require membrane fusion, and the receptor–ligand binding triggers downstream signaling that is sufficient to induce physiological changes in the target cell [17]. The second mechanism includes the fusion of EVs with target cells, which results in actual transfer of the EV and its cargo into the target cell [18]. Several pathways for EV uptake are known, and different cell or EV types may use different mechanisms, including phagocytosis, pinocytosis, receptor-mediated endocytosis, and membrane fusion [19]. After EV uptake, the EVs are degraded, and their content is released into the recipient cytosol [20].

Hypoxia changes the content of EVs; as such, hypoxic cancer-cell-derived EVs have been linked to the induction of cellular effects in distant recipient cells, as well as the transfer of hypoxia tolerance to other (non-hypoxic) cells [21,22]. Hypoxia also affects EV biogenesis through regulation of RAB GTPases that are involved in EV secretion [23,24]. For example, hypoxia-induced STAT3 regulates RAB proteins, promotes the release of EVs by ovarian cancer cells, and results in a more aggressive cancer phenotype [24]. Although the exact mechanisms that control EV biogenesis in response to hypoxia remain to be clarified, recent evidence indicates that they are distinct from regular EV biogenesis pathways. This is illustrated by the difference in size and the dependence on GABA_A_ receptor-associated protein-like 1 (GABARAPL1) [22].

In this review, we discuss the current knowledge about the role of hypoxic cancer-cell-derived EVs in immunosuppression (Figure 1). In addition, other factors (i.e., miRNA and proteins) known to be enriched in hypoxic EVs are described. Although no direct evidence for their role in EV-mediated immune regulation exists, these factors have been shown to affect immune cells’ function, maturation, and migration (Figure 2). For these factors, we describe their immunomodulatory function per immune cell type, as they might contribute to the immunosuppressive phenotype of hypoxic cancer-cell-derived EVs.

## 2. Neutrophils

Neutrophils are the most abundant population of white blood cells in the human circulatory system, reflecting their role as first responders during infection or tissue injury. Elevated neutrophil counts are observed in the peripheral blood and tumors of patients with advanced cancers. In these patients, both tumor-supportive and tumor-suppressive neutrophil subsets have been described (Table 1) [25,26]. The direction of the tumor-associated neutrophil function is dependent on the types of soluble mediators produced by cancer cells. For example, in a transforming growth factor β (TGF-β)-rich environment, neutrophils typically display a tumor-promoting phenotype, whereas they acquire antitumor properties in the presence of interferon β (IFN-β) [25]. To date, no direct effect of hypoxic EVs on neutrophil function has yet been described. Nevertheless, hypoxic cancer-derived EVs have been shown to be enriched in factors known to play a role in neutrophil chemotaxis, including CXCL8 (IL-8) and *MIR-451* (Figure 2A). CXCL8 is a pro-inflammatory cytokine that is enriched in EVs of hypoxic glioma cells [27]. It mediates the recruitment of leukocytes—including neutrophils and myeloid-derived suppressor cells (MDSCs)—to tumors through binding the receptors CXCR1 and/or CXCR2 [28]. Moreover, through activation of MAPK signaling, CXCL8 signaling inhibits neutrophil apoptosis and enhances proliferation [29]. An opposing role has been observed for *MIR-451*, which inhibits the p38 MAPK signaling pathway, suppressing neutrophil chemotaxis [30]. Given the known diversity in pro- or antitumor neutrophil subpopulations and the varying effects of the different hypoxic-EV-associated factors, further research is warranted to clarify the effects of hypoxic tumor-derived EVs on the influx of neutrophils with either antitumor or tumor-suppressive properties.

## 3. Macrophages

Macrophages, as phagocytosing cells of the innate immune system, play an important role in maintaining tissue homeostasis. A variety of macrophage phenotypes have been identified, each with different effector functions [31]. Importantly, tumor-associated macrophages (TAMs) are functionally distinct from the normal and tissue-resident populations [32]. A commonly used paradigm categorizes macrophages into pro-inflammatory (i.e., antitumor) and an anti-inflammatory (i.e., tumor-promoting) phenotypes, also known as the ‘M1’ and ‘M2’ subtypes, respectively [33,34]. These macrophage subtypes are characterized by different metabolic profiles, reflecting their functions [35,36]. M1 activated macrophages perform aerobic glycolysis, with reduced activity of their respiratory chain which, in turn, allows the production of reactive oxygen species (ROS) to support their pro-inflammatory function [35,36]. Conversely, M2 macrophages rely mostly on oxidative metabolism for their energy production [35,36]. Forcing macrophage metabolism towards either glycolysis or oxidative metabolism also skews their phenotype towards M1 or M2, respectively [35]. TAMs share many phenotypic and functional characteristics with the M2 subtype, as they facilitate tumor cells’ proliferation, invasion, intravasation, and metastatic dissemination. Moreover, TAM-derived factors have proangiogenic and immunosuppressive effects [37]. In vitro stimulation can enrich cells with these phenotypes on either side of the spectrum. However, in vivo macrophage phenotypes are more dynamic and complex [38,39]. Accumulating evidence suggests that tumor-derived EVs mediate M2 polarization of macrophages, and thereby contribute to cancer progression [40,41].

EVs derived from hypoxic melanoma cells are enriched in a range of pro-tumorigenic and immunosuppressive factors, including TGF-β1, TGF-β2, TGF-β3, macrophage migration inhibitory factor (MIF), and ferritin heavy/light chain (FTH/FTL) (Table 2) [42]. Stimulation with these hypoxic EVs shifts bone-marrow-derived macrophages (BMDM) towards the anti-inflammatory M2 phenotype, as characterized by the upregulation of arginase 1 *(Arg1*), *Ym1* (chitinase 3-like 3), and *Fizz1* [42]. In addition, they increase the expression of the TAM-associated genes *Cox-2*, *Pges-1*, and *ll-6*, which play important roles in immunosuppression and tumor growth [42]. A similar shift in macrophage phenotype is observed when macrophages are exposed to EVs derived from hypoxic glioblastoma, lung, pancreatic, endometrial, and epithelial ovarian cancer cells [43,44,45,46,47,48]. In turn, these macrophages increase cancer cells’ proliferation, migration, invasion, and angiogenesis in vitro [44,45,46,47,48]. Furthermore, these EVs induce the infiltration of M2 macrophages into the tumor, and mice develop a more aggressive disease with increased metastatic burden and shorter survival times [42,44,45,46,48]. These effects have been attributed to the transfer of hypoxia-upregulated miRNAs via EVs. Transfer of *MIR-1246* targets the expression of TERF2IP in macrophages which, in turn, activates signal transducer and activator of transcription 3 (STAT3) signaling and inhibits nuclear factor kappa B (NF-κB), subsequently skewing the cells to the M2 phenotype [44]. In addition, *MIR-103a* and *MIR-301a-3p* target the expression of phosphatase and tensin homolog (PTEN), subsequently activating STAT3, RAC-alpha serine/threonine protein kinase (AKT), and PI3Kγ, thereby also skewing macrophages to the tumor-promoting M2 phenotype [45,46]. In summary, hypoxic EVs from various types of cancer, and their specific contents, are able to influence macrophages in the tumor microenvironment, skewing the macrophage compartment towards a tumor-supportive role (Figure 1A).

Other factors that can alter macrophage behavior include *MIR-21*, *MIR-23*, *MIR-92A*, *MIR-127*, *MIR-135*, *MIR-210*, *MIR-494*, *MIR-1246*, and carbonic anhydrase IX (CAIX). These factors are enriched in hypoxic EVs (Table 2), and their effects may therefore be mediated through EV secretion, although this has not been reported directly (Figure 2B). CAIX is an enzyme that plays an important role in the acidification of the extracellular environment [49]. This extracellular acidosis is known to reduce the function of various immune cells, including T cells, natural killer (NK) cells, and dendritic cells [50]. In macrophages, this acidification causes metabolic reprograming, pushing them towards the oxidation of fatty acids [51]. This, in turn, skews them towards the tumor-promoting M2 phenotype [51]. *MIR-21*, *MIR-23*, and *MIR-494* affect macrophage behavior by downregulating the expression of PTEN [52,53,54,55]. The subsequent suppression of NF-κB and AKT activity pushes the macrophages towards an M2 phenotype, with reduced production of tumor necrosis factor alpha (TNF-α) and IL-1β, and increased production of IL-10, CD206, and PD-L1 [52,53,54,55]. In addition, these macrophages induce T-cell apoptosis [54]. Several other factors enriched in EVs derived from hypoxic cancer cells have the potential to alter the secretome of macrophages to support a pro-tumor phenotype. Macrophage-derived pro-inflammatory factors, such as TNF-α and ROS, may be suppressed by EV-derived *MIR-135* [56]. Tumor-EV-derived *MIR-92A* has also been shown to alter macrophages to increase their production of IL-6 which, in turn, stimulates cancer cells’ proliferation, migration and invasion [57]. For other factors, the direction of stimulation is less clear, and may be dependent on the context or additional signaling. For example, *MIR-127* can target the expression of CD64 and TRAF1 to support an anti-inflammatory phenotype in macrophages with decreased production of pro-inflammatory TNF-α, IL-1β, and IL-6 [58,59]. On the other hand, *MIR-127* has also been associated with M1 macrophage polarization via targeting of BCL-6 and subsequent regulation of DUSP1 expression and JNK activity [60]. Further research is thus required to clarify its role in macrophage polarization.

In addition to altering macrophage function as described above, induction of macrophage cell death or prevention of full macrophage differentiation are alternative mechanisms that can suppress macrophage-mediated antitumor reactions. Under hypoxia, cancer cells secrete increased levels of *MIR-210* and *MIR-1246* via their EVs [44,61]. *MIR-210* targets DECR1 and promotes mitochondrial dysfunction and necroptosis in macrophages [62]. *MIR-1246* targets the expression of caveolin-1, which is required for the differentiation of monocytes into macrophages [63,64]. Downregulation of caveolin-1 in macrophages increases their VEGF-A/VEGFR1 signaling activity and downstream expression of matrix metalloproteinase 9 (MMP9) and colony-stimulating factor 1 (CSF1), thereby facilitating angiogenesis and metastasis formation [65].

In addition to miRNAs, hypoxic cancer-cell-derived EVs are enriched in pyruvate kinase M2 (PKM2). PKM2 is a key enzyme in the glycolytic pathway whose functionality changes depending on its polymerization [66]. As a tetramer, it has a high affinity for its substrate (phosphoenolpyruvate), and stimulates glycolysis [66]. However, as a dimer, it has lower enzymatic activity, but is able to translocate to the nucleus, where it acts as a protein kinase of several transcription factors [66]. It has been described that EV-associated PKM2 can determine macrophage cell fates. THP1 monocytes cultured with normoxic EVs containing PKM2 display increased glycolysis through regulation of hypoxia-inducible factor 1α (HIF-1α), which induces the differentiation of monocytes to macrophages [67]. Moreover, macrophage differentiation is further controlled by the translocation of PKM2 to the nuclei of monocytes, where it phosphorylates STAT3 [67]. Phosphorylated STAT3 is required for the production of the differentiation-associated transcription factors MAFB, C-MAF, and EGR-1. In these macrophages, M1 markers are downregulated and M2 markers are upregulated [67]. Similarly, PKM2 was shown to inhibit LPS-induced pro-inflammatory M1 macrophage polarization, while promoting M2 macrophage traits through the inhibition of IL-1β production and boosting the production of IL-10 [68]. The increased levels of PKM2 found in the EVs of hypoxic cancer cells might further enhance the abovementioned effects, and thereby induce a stronger immunosuppressive phenotype in recipient monocytes compared to normoxic EVs. In conclusion, hypoxic cancer-cell-derived EVs are enriched in numerous factors able to affect macrophage differentiation, polarization, and migration and, as such, contribute to cancer progression (Figure 2B).

**Table 2 cancers-14-04001-t002:** EV-enriched factors and their effects on macrophages.

Factor	Effect on Macrophages	Proven to Be EV-Mediated? ^1^
TGF-β1 TGF-β2 TGF-β3 [42]	Induces anti-inflammatory M2 phenotype with expression of TAM-associated genes.	Yes
CAIX [49,51]	Involved in extracellular acidification which, in turn, causes a metabolic switch in macrophages, inducing the M2 phenotype	No
MIF [42]	Induces anti-inflammatory M2 phenotype with expression of TAM-associated genes.	Yes
FTH/FTL [42]	Induces anti-inflammatory M2 phenotype with expression of TAM-associated genes.	Yes
*MIR-1246* [44,63,64,65]	Induces anti-inflammatory M2 phenotype via NF-κB inhibition. Limits differentiation of monocytes into macrophages via reduced caveolin-1 expression. Increases macrophage-mediated angiogenesis and metastasis formation.	Yes
*MIR-103a* [45]	Induces anti-inflammatory M2 phenotype via reduced PTEN expression.	Yes
*MIR-301a-3P* [46]	Induces anti-inflammatory M2 phenotype via reduced PTEN expression.	Yes
*MIR-21* [52,53]	Induces anti-inflammatory M2 phenotype via reduced PTEN expression.	No
*MIR-23* [54]	Induces anti-inflammatory M2 phenotype via reduced PTEN expression.	No
*MIR-494* [55]	Induces anti-inflammatory M2 phenotype via reduced PTEN expression.	No
*MIR-135* [56]	Reduces production of pro-inflammatory factors TNF-α and ROS.	No
*MIR-92a* [57]	Enhances production of tumor-supportive IL-6.	No
*MIR-127* [58,59,60]	Induces anti-inflammatory M2 phenotype via reduced CD64 and Traf1 expression. Induces pro-inflammatory M1 phenotype via reduced BCL-6 expression.	No
*MIR-210* [62]	Induces necroptosis via reduced DECR1 expression.	No
PKM2 [67,68]	Induces anti-inflammatory M2 phenotype via STAT3 phosphorylation.	No

^1^ Yes or no refers to the current knowledge regarding the respective hypoxic-EV-enriched proteins or miRNAs. Yes: a direct EV-mediated effect on the immune cell has been identified. No: no direct EV-mediated effects have yet been described, but the miRNA or protein has known immunomodulatory functions.

## 4. Myeloid-Derived Suppressor Cells

MDSCs are a heterogeneous group of immature myeloid immune cells that can increase in number during pathological conditions such as chronic inflammation and cancer [69]. They have attracted great scientific interest due to their ability to suppress T-cell-mediated tumor clearance and foster tumor progression through the production of inducible nitric oxide synthase (iNOS) and arginase, among others [70]. MDSCs have high plasticity, and can further differentiate into dendritic cells, tumor-associated macrophages, and granulocytes within tumor environments [71]. In addition to their T-cell-suppressive function, MDSCs have a pro-angiogenic secretory profile, and stimulate cancer-supporting inflammation [72]. It has been shown that cancer-derived EVs can stimulate the development of MDSCs as well as their tumor-supportive effector functions [73,74]. Interestingly, vesicles derived from hypoxic glioma cells have a greater MDSC-inducing capacity compared to normoxic EVs, associated with increased production of immunosuppressive factors such as TGF-β, IL-10, nitric oxide (NO), ROS, and arginase [73,74]. Consequently, these MDSCs inhibit CD8^+^ T-cell proliferation more efficiently compared to MDSCs stimulated with normoxic control EVs (Table 3) [73,74]. These immunosuppressive effects of hypoxic glioma EVs have been attributed to *MIR-10a* and *MIR-21*, which reduce the expression of RAR-related orphan receptor alpha (*Rora*) and *Pten*, respectively [73]. The subsequent translocation of NF-κB into the nucleus and increased AKT activation induce MDSC activation [73]. In addition, increased levels of *MIR-29a* and *MIR-92a* within the hypoxic glioma-derived EVs decrease the expression of the high-mobility group box transcription factor 1 (*Hbp1*) and protein kinase CAMP-dependent type I regulatory subunit alpha (*Prkar1α*) genes in MDSCs, respectively [74]. *Hbp1* inhibits cell-cycle progression in the G1 phase, while *Prkar1α* is part of the PKA/p-STAT3 pathway, where it regulates the production of ROS, TGF-β, and IL-10 [74]. Targeting these genes through miRNAs results in increased proliferation, and increases the production of immunosuppressive factors by MDSCs [74]. In conclusion, tumor-derived EVs—especially hypoxic populations—play important roles in the development and function of pathological MDSCs, which protect tumor cells from antitumor immune reactions (Figure 1B).

In addition, various other factors known to modulate MDSC function are increased in EVs derived from hypoxic cancer cells, including *MIR-210*, *MIR-494*, C-C motif chemokine ligand (CCL)2, and insulin-like growth-factor-binding protein 3 (IGFBP-3) (Table 3) [75,76,77]. *MIR-210* enhances the immunosuppressive effects of MDSCs via increases in *Arg1* expression, ARG activity, and NO production [75]. Consequently, overexpression of *MIR*-*210* in MDSCs results in increased tumor growth due to increased T-cell suppression [75]. *MIR-494* reduces PTEN expression in MDSCs, subsequently activating AKT, NF-κB, and mammalian target of rapamycin (mTOR) which, in turn, potentiate their immunosuppressive function, thereby supporting tumor growth [76]. Murine models have shown that CCL2 stimulates the production of ROS and NO by MDSCs, which subsequently suppress T-cell proliferation [78]. In humans, stimulation of CCL2 production in monocytes results in enhanced differentiation towards MDSCs [79]. IGFBP-3, enriched in hypoxic cancer-cell-derived EVs, is a factor known to induce a subpopulation of CD38^high^ MDSC cells with an even more immature phenotype compared to MDSCs lacking CD38. Subsequently, these CD38^high^ MDSCs have a greater capacity to inhibit activated T cells [80]. Transfer of these factors from hypoxic cancer cells via EVs may induce the development of MDSCs with a great T-cell-suppressive capacity, thereby supporting tumor growth (Figure 2C).

**Table 3 cancers-14-04001-t003:** EV-enriched factors and their effects on MDSCs.

Factor	Effect on MDSCs	Proven to Be EV-Mediated? ^1^
*MIR-10a* [73]	Potentiates MDSC function via reduced *Rora* expression.	Yes
*MIR-21* [73]	Potentiates MDSC function via reduced *Pten* expression.	Yes
*MIR-29a* [74]	Increases MDSC proliferation via reduced *Hbp1* expression.	Yes
*MIR-92a* [74]	Increases the production of immunosuppressive factors by MDSCs via reduced Prkar1α expression.	Yes
*MIR-210* [75]	Enhances the immunosuppressive effects of MDSCs via increased ARG activity and NO production.	No
*MIR-494* [76]	Stimulates MDSCs’ immunosuppressive effects via targeting of PTEN.	No
CCL2 [79]	Stimulates immunosuppressive effects of MDSCs.	No
IGFBP-3 [80]	Induces a more efficient CD38^high^ MDSC population.	No

^1^ Yes or no refers to the current knowledge regarding the respective hypoxic-EV-enriched proteins or miRNAs. Yes: a direct EV-mediated effect on the immune cell has been identified. No: no direct EV-mediated effects have yet been described, but the miRNA or protein has known immunomodulatory functions.

## 5. Dendritic Cells

Conventional antigen-presenting dendritic cells (cDCs) are a crucial bridge between the innate and adaptive immune responses. DCs originate from hematopoietic stem cells. These progenitor cells differentiate into immature DCs, which infiltrate peripheral tissues to sample antigens [81]. Upon encountering a foreign antigen, immature DCs transition into mature DCs [82]. Three major subtypes of DCs have been described: The cDC1s are able to cross-present antigens to CD8^+^ T cells, resulting in the activation of the cytotoxic T cells and, subsequently, the activation of Th1 CD4^+^ T cells [83,84]. The cDC2s are preferentially involved in the activation of CD4^+^ T cells [85]. Therefore, successful DC maturation upon encountering an antigen is important, since DC-mediated priming of T cells is required for unleashing antitumor T-cell responses. The plasmacytoid DCs (pDCs) are the third and rarest subset of DCs; pDCs differ from the other cDC subsets in terms of development and antigen-presenting function. However, the production of high levels of type I IFNs by pDCs, which is observed in response to viral infections, may also play a role in antitumor immunity [86,87]. The currently available studies mainly demonstrate the effects of hypoxia on the cDC subsets.

Tumor-derived EVs exert immunosuppressive effects by reducing the ability of DCs to orchestrate the adaptive antitumor immune response (Table 4). Breast-cancer-derived EVs block the differentiation of myeloid precursor cells into DCs, and this effect is at least partially mediated via the induction of IL-6 in recipient myeloid precursor cells/monocytes [88]. Moreover, melanoma-derived EVs suppress DCs’ maturation in vitro, as indicated by impaired expression of CD83 and CD86, as well as decreased expression of chemokines inducing Th1 polarization. Although various factors known to affect DC differentiation have been identified in these EVs, the exact cargo responsible for the observed effects remains to be elucidated [89]. Furthermore, the contribution of hypoxia to this EV-mediated suppression of DC differentiation is currently unknown.

Hypoxic cancer-cell-derived EVs are enriched in *MIR-301* and *MIR-451*. Although no direct evidence for their role in EV-mediated immunosuppression exists, both miRNAs have been shown to reduce the production of pro-inflammatory cytokines such as IL-6, IL-12, CCL3/MIP1a, CCL5/RANTES, and TNF-α in dendritic cells, which could result in less activation of Th1 adaptive antitumor immune responses [90,91]. Furthermore, expression of *MIR-301* in DCs represses the release of IFN-γ from DC-primed CD8^+^ and CD4^+^ responder cells [90]. In addition, increased levels of CCL2 in hypoxic EVs could affect the differentiation of monocytes into DCs [92]. Supplementation of a standard DC differentiation cocktail (GM-CSF and IL-4) with CCL2 resulted in lower production of IL-12 which, in turn, hampered effective IFN-γ-mediated T-cell cytotoxicity [92]. Collectively, these data show how hypoxic cancer-derived EVs, via transfer of *MIR-301*, *MIR-451*, and CCL2, may be able to interfere with dendritic cells’ functioning, resulting in reduced antitumor immune reactions (Figure 2D).

## 6. NK Cells

NK cells are part of the innate immune system, and are among the first responders in the inflammatory cascade. They have the capacity to recognize and kill infected, neoplastic, and/or malignantly transformed cells. NK cells recognize a potential target cell through interaction between their membrane-associated receptors and the activating or inhibitory ligands expressed by the target cell [93]. If the signaling through activating NK cell receptors is higher compared to the signaling through inhibitory receptor–ligand interactions, NK cells’ effector functions are triggered. These include NK cell degranulation and release of their toxic granules containing perforin and granzymes, killing of target cells via death receptors, and the secretion of IFN-γ. NK cells are controlled by a very broad panel of inhibitory and activating receptors, although the most important group of inhibitory ligands are the major histocompatibility complex class I (MHC-1) molecules, which are expressed in virtually every healthy cell [93].

Cancer cells have been shown to evade NK-mediated killing. In several cancer models, EVs have been shown to regulate NK cells’ antitumor effector functions and, depending on the exact makeup of the vesicles, this results in either stimulation or an inhibition of NK cells’ antitumor functions, as reviewed in [94]. Moreover, tumor hypoxia has been identified as one of the important factors contributing to tumor cells’ resistance to NK cell killing [95]—presumably because hypoxia enhances tumor cell autophagy, which in breast cancer cells has been shown to reduce NK cell killing via the breakdown of NK-secreted granzymes in the tumor cells [96]. Although the impact of (hypoxic) EVs on NK cells is only very poorly understood, one study showed that hypoxic lung-cancer-cell-derived EVs may reduce NK-cell-mediated cytotoxicity via transfer of the immunosuppressive cytokine TGF-β, which decreases IFN-γ expression and secretion, and decreases surface expression of the activating receptor NKG2D. Additionally, these EVs transfer *MIR-23A*, which directly targets the expression of degranulation-associated CD107a, resulting in a lower proportion of degranulating NK cells when encountering a target cell (Table 5) [97] (Figure 1C).

## 7. T Cells

T cells are part of the adaptive arm of the immune response, and express either the CD4 or CD8 glycoprotein, which roughly subdivides them into helper T cells and cytotoxic T cells. Upon activation, these T cells undergo clonal expansion to generate a large pool of antigen-specific lymphocytes, which then further differentiate into specialized effector T cells [98]. Naïve CD4^+^ T-helper cells can further differentiate into subsets, which are distinguishable based on their surface marker expression and cytokine profile. The principal subsets of CD4^+^ T cells that play key roles in cancer development are T-helper (Th)1, Th2, Th17, and regulatory T (Treg) cells [99]. These CD4^+^ T-cell subsets can play opposing roles in cancer progression. Th1 cells secrete pro-inflammatory type 1 cytokines (including IFN-γ, TNF-α, CCL2, and CCL3), which support antitumor immune reactions and CD8^+^ T-cell activation. In contrast, Th2 cells secrete type 2 cytokines (such as IL-4, -5, and -13), which promote type II immune responses (e.g., M2 macrophage polarization), tissue remodeling, and tumor growth. The role of Th17 in tumor progression remains controversial, with reports of both tumor-promoting angiogenic effects and antitumor immune reactions [99,100]. Lastly, Treg cells suppress immune responses by producing cytokines (e.g., IL-10, TGF-β) that inhibit the activation of lymphocytes, dendritic cells, and macrophages [98]. Furthermore, they may render antigen-presenting cells unable to provide the co-stimulatory signals needed for T-cell activation, thus suppressing antitumor immune reactions [98]. Cancer cells modulate these different T-cell subsets in various ways to support tumor growth, including via the release of EVs (Table 6) [101,102]. EVs derived from nasopharyngeal carcinoma cells have been shown to suppress the proliferation of CD4^+^ and CD8^+^ T-cell subsets and inhibit the differentiation towards Th1 and Th17 subtypes [103]. In addition, they increase the percentage of FOXP3-positive Treg cells, skewing the TME towards tumor-supportive immunosuppression [103]. These effects can be attributed to vesicular *MIR-24-3p*, and are more pronounced in hypoxic conditions [103] (Figure 1D).

### 7.1. Hypoxic EVs May Limit Differentiation and Regulate Polarization toward Tumor-Supportive Subtypes via Transfer of Factors with Known Immunomodulatory Roles

Hypoxic EVs are enriched in a number of factors for which immunomodulatory roles have been described independent of EV-mediated communication. Factors that can alter T-cell differentiation and maturation include *MIR-125*, *MIR-210*, and thrombospondin-1 (TSP-1) [61,73,104]. *MIR-125* suppresses CD4^+^ T-cell differentiation and maintains a naïve T-cell state by decreasing the expression of *Ifn-γ*, *TNF-α*, *IL-2Rβ*, *IL-10Rα*, PR domain zinc finger protein 1 (*BLIMP-1)*, *Stat3*, and *Il-13* [104,105,106]. Furthermore, *MIR-125*-mediated suppression of STAT3, IL-13, and IFN-γ stabilizes Treg lineage commitment, which contributes to the formation of an immunosuppressive TME [105]. Similarly, TSP-1 induces the differentiation of immunosuppressive Treg cells [107]. HIF-1α plays an important role in the polarization of T cells and regulates the balance between Th17 and Treg polarization [108,109]. *MIR-210*, upregulated in the EVs from hypoxic cancer cells, reduces the expression of HIF-1α, which subsequently suppresses Th17 differentiation and reduces inflammation [110]. Conversely, PKM2 promotes Th17 differentiation through STAT3 activation [111]. However, as mentioned above, the role of Th17 cells in cancer immunity is still paradoxical, and further research is necessary to understand their contribution to pro- versus antitumor immune reactions. In summary, hypoxic EVs can contribute to the immunosuppressive TME by interfering with the development of functional T cells, or by skewing their polarization into tumor-supportive subtypes.

### 7.2. Hypoxic EVs May Decrease the Abundance of T Cells (Infiltration and Proliferation) in the Tumor and Limit Antitumor Immunity via Transfer of Factors with Known Immunomodulatory Roles

Reducing T-cell-mediated cytotoxic efficacy yields efficient suppression of antitumor immune reactions. This reduced cytotoxic efficacy could be induced through hypoxic-EV-mediated communication between tumor cells and target cells. As such, *MIR-23*, TGF-β, and CAIX are enriched in hypoxic EVs, and these factors can limit cytotoxic T-cell effector function [97,112,113,114,115,116]. *MIR-23* blunts the expression of multiple key cytotoxic T-lymphocyte (CTL) effector molecules, including IFN-*γ*, thereby suppressing T-cell-mediated cytotoxicity and, subsequently, accelerating tumor progression and increasing the tumor burden [117]. These effects are mediated by suppression of BLIMP-1—a key regulator in the activation of cytotoxic T-cell immune reactions [117,118,119]. TGF-β increases the expression of *MIR-23* in CTLs, thereby further contributing to immunosuppression [97,114,117]. CAIX, as described above, plays an important role in extracellular acidification [49]. In T cells, this acidification inhibits glycolysis which, in turn, induces anergy, with impaired proliferation and reduced cytolytic activity and cytokine release [120,121,122]. In addition, extracellular acidification has also been postulated to suppress T-cell differentiation through the suppression of mTORC1 [50]. mTORC1 plays an important role in the differentiation of effector T cells, and its inhibition skews their differentiation towards immunosuppressive Treg cells [123,124]. Similarly, high CAIX expression in tumors is correlated with high infiltration of FOXP3^+^ Treg cells [125].

In addition to a reduction in T-cell-mediated killing efficacy, suppression of T-cell proliferation also severely reduces the efficacy of the immune response. MMP9 and *Let-7a* are enriched in hypoxic EVs, and have the capability to inhibit T-cell proliferation [42,126]. MMP9 mediates shedding of the IL-2 receptor-α from the surface of T cells, thereby preventing CD4^+^ CD8^+^ T-cell proliferation [127]. *Let-7a* reduces both the infiltration of T cells into the tumor and CD3^+^ T-cell proliferation [128,129]. Moreover, *Let-7a* reduces the secretion of IFN-γ by T cells via decreased STAT3 expression [129]. This indicates that, in addition to the crude number of cells present, EV-associated factor *Let-7a* inhibits immune cells’ effector function, presenting another approach via which EV-associated factors can modulate tumor immunology.

Alternatively, various factors enriched in EVs of hypoxic cancer cells have been shown to limit the infiltration of immune cells into the tumor, including IGFBP-3, TSP-1, and a disintegrin and metalloprotease with thrombospondin motif 1 (ADAMTS1) [27,113]. In mice with mammary tumors deficient in IGFBP-3, the intratumoral gene expression levels of *Ifn-γ*, *Cd8*, and *Tnf-α* were elevated, indicating a higher infiltration of CD8+ cytotoxic T cells into the tumor [130]. In triple-negative breast cancer, TSP-1 expression was observed to be inversely correlated with the infiltration of CD8+ lymphocytes. Moreover, this was recapitulated by TSP-1 knockdown in the 4T1 metastatic mouse model [131]. Transfer of IGFBP-3 via (hypoxic) cancer-derived EVs might thus suppress immune infiltration into the tumor and, as such, stimulate tumor growth. Another hypoxia-enriched factor in EVs that affects both the infiltration and effector function of T cells is ADAMTS1—an extracellular protease for which both anti- and pro-tumorigenic functions have been described [132]. Knockout of ADAMTS1 in mammary and melanoma cancer models resulted in reduced primary and secondary tumor burden, which correlated with increased infiltration of cytotoxic lymphocytes and increased expression of genes involved in antitumor immune reactions [133,134]. However, this response may be dependent on the expression and abundance of ADAMTS1 substrates such as the matrix proteoglycan versican (VCAN) in the TME. Intact VCAN triggers the secretion of cytotoxic T-cell-inhibitory cytokines by antigen-presenting cells, whereas its proteolytic product versikine triggers the secretion of cytotoxic T-cell-activating cytokines. However, the balance between ADAMTS1′s pro- and anti-inflammatory effects is currently not fully understood, leaving the exact immune response in these settings unknown [135]. In conclusion, hypoxic cancer-cell-derived EVs may play a role in the evasion of antitumor immune reactions by reducing T-cell functionality and by limiting the proliferation and infiltration of cytotoxic T cells, while stimulating the recruitment of immunosuppressive regulatory T cells (Figure 2F).

**Table 6 cancers-14-04001-t006:** EV-enriched factors and their effects on T cells.

Factor	Effect on T Cells	Proven to Be EV-Mediated? ^1^
*MIR-23* [117,118,119]	Suppresses T-cell-mediated cytotoxicity by reducing the expression of BLIMP-1.	No
*MIR-24-3p* [103]	Reduces proliferation of CD4^+^ and CD8^+^ T cells. Inhibits differentiation towards Th1 and Th17 subtypes. Increases the FOXP3^+^ Treg cell population.	Yes
*MIR-125* [104,105,106]	Maintains a naïve T-cell state by decreasing the expression of *Ifn-γ*, *TNF-α*, *IL-2Rβ*, *IL-10Rα*, *BLIMP-1*, *Stat3*, and *Il-13*. Stabilizes Treg lineage commitment via suppression of *STAT3*, *Il-13*, and *Ifn-γ* expression.	No
*MIR-210* [110]	Suppresses Th17 differentiation and reduces inflammation via reduced HIF-1α expression.	No
*Let-7a* [128,129]	Reduces T-cell proliferation and infiltration. Reduces IFN-γ secretion via reduced STAT3 expression.	No
ADAMTS1 [133,134]	Negatively influences the infiltration of cytotoxic lymphocytes and the expression of antitumor immune gene profiles.	No
CAIX [120,121,122]	Induces lymphocyte anergy via extracellular acidification. Hinders T-cell differentiation via extracellular acidification.	No
MMP9 [127]	Prevents T-cell proliferation via shedding of IL-2 receptor-α.	No
IGFBP-3 [130]	Suppresses immune infiltration into the tumor.	No
TSP-1 [131]	Reduces the infiltration of CD8^+^ lymphocytes. Decreases inflammatory IFN-γ signaling via activation of TGF-β. Induces the differentiation of Treg cells.	No
PKM2 [111]	Promotes Th17 differentiation through STAT3 activation.	No

^1^ Yes or no refers to the current knowledge regarding the respective hypoxic EV-enriched proteins or miRNAs. Yes: a direct EV-mediated effect on the immune cell has been identified. No: no direct EV-mediated effects have yet been described, but the miRNA or protein has known immunomodulatory functions.

## 8. NKT Cells

NKT cells are a developmentally and functionally distinct lineage of T cells that recognize lipid antigens, such as glycolipids and glycerol lipids, presented by CD1d—an MHC class Ib molecule. Different NKT cell subtypes with opposing functions have been described. Type I NKT cells have effective antitumor effector functions. On the other hand, type II NKT cells are regulators of immunosuppression [136]. Hypoxic EV-associated factors may regulate the skewing of NKT-cell subsets (Table 7). While there are no studies directly investigating the effects of hypoxic EVs on NKT cells, it is known that hypoxic cancer cells increase their EV-mediated secretion of *MIR-92A* [126]. In another study, *MIR-92A* has been shown to induce the expression of IL-6 and IL-10 in NKT cells, reducing their direct antitumor effects and leading to NKT-mediated suppression of CD8^+^ T-cell proliferation [137]. As such, increased loading of *MIR-92A* into EVs under hypoxic conditions may promote tumor growth via the induction of immunosuppressive cells (Figure 2F).

## 9. B Cells

B cells play a key role in immunological memory of infectious diseases as long-lived and antibody-producing plasma cells or memory B cells that are capable of responding to reinfections. High abundances of tumor-infiltrating B cells in solid cancers are commonly found [138]. However, their exact role in the TME is not yet clear, as both tumor-growth-promoting and antitumor immune reactions have been described [138]. Several factors known to suppress B-cell functioning are enriched in the EVs of hypoxic cancer cells, including *MIR-210* and *MIR*-125 (Table 8) [61,73]. *MIR-210* acts as a negative regulator of B-cell-mediated immune responses, and fine-tunes the balance between pathogen clearance and autoimmunity [139]. Its overexpression results in impaired B-cell proliferation and antibody production [139]. In addition, *MIR-125* prevents B cells’ maturation and release from the bone marrow [73,140]. As such, delivery of *MIR-210* and *MIR-125* to B cells via EVs may suppress B-cell-mediated immune reactions (Figure 2G). However, as the role of B cells in tumor immunity is yet to be fully elucidated, further research is required [138].

## 10. Immune Stimulation by Hypoxia-Upregulated Factors in EVs

Various factors enriched in hypoxic cancer-cell-derived EVs also stimulate the immune system, and could therefore induce antitumor immune reactions (Table 9). *MIR-181* enhances the development of NK cells by downregulating the expression of nemo-like kinase (NLK)—a negative regulator of the Notch signaling pathway [141,142]. Activation of the Notch signaling pathway stimulates both early and late phases of NK cells’ development [143,144,145]. In addition, *MIR-155* targets the expression of SH2 domain-containing inositol 5′-phosphatase 1 (SHIP1) in NK cells [146]. SHIP1 is a negative regulator of pro-inflammatory IFN-γ production, and its downregulation by *MIR-155* could therefore enhance antitumor immune responses [146]. Furthermore, *MIR-155* is required to elicit an effective antitumor immune response in macrophages, as it targets the expression of IL-13Rα1. This reduces the capacity to respond to M2-inducing signals, and pushes the macrophages towards the M1 phenotype via targeting of interferon regulatory factor 4 (*Irf4*) [147,148,149,150,151,152]. In CD8^+^ T cells, *MIR-155* plays an important role in antitumor immunity, as it regulates the expression of several critical activation and effector genes, including *Ifn-γ*, *Granzyme B*, *Perforin*, *Klrg*, *Cd62l*, *Ctla4*, and *Pdcd1* [153]. Furthermore, *MIR-155* reduces *Socs1* expression, which modulates CD8^+^ T cells’ responsiveness to IL-2, IL-7, and IL-15 stimulation [154]. Taken together, these observations show the dual nature of hypoxia-upregulated factors in EVs in both immunosuppression and immune stimulation. These contradictory roles could be explained by the existence of different EV subpopulations, which contain different protein and miRNA cargoes [155,156]. Consequently, these different subpopulations might be selectively taken up by specific recipient cells; as such, one batch of EVs can elicit different effects, depending on the receptor cells. The complexity and heterogeneity of EVs warrants further investigation into the different subpopulations of EVs and their pathological roles, as a more thorough understanding of their biology will pave the way towards new therapeutic strategies and biomarker development.

## 11. Importance of EV Isolation Methodology and Experimental Setup

Aiming to increase the reproducibility and reliability of published EV results, the International Society for Extracellular Vesicles has established the Minimal Information for Studies of Extracellular Vesicles (MISEV) guidelines [16]. These guidelines provide recommendations on EV isolation, characterization, and functional studies, and emphasize the importance of reporting requirements specific to the EV field [16]. Despite these efforts, many studies still base their conclusions on experimental procedures that may not be optimal for their purpose. One of the main problems is the use of EV isolation techniques that disregard EVs’ purity and morphology/functionality. For example, ultracentrifugation—one of the most widely used methods to isolate EVs—is known to suffer from various drawbacks, such as the incomplete sedimentation of EVs, co-isolation of non-EV protein/RNA impurities and aggregates, and possible damage to EVs [157,158,159,160,161]. The same holds true for precipitation-based isolation strategies [157]. The impacts of the different isolation methodologies on the obtained results have been extensively described previously [157,158,159,160,161,162,163,164]. In addition, the field of hypoxia research also suffers from diversity in experimental setup, with different oxygen concentrations and chemical models such as CoCl_2_ being applied to simulate tumor hypoxia. As results obtained using different EV isolation methods and experimental setups may introduce variability, we summarize the details of the discussed studies in Table 10 and Table 11.

## 12. Conclusions

In conclusion, tumors use EVs to communicate with immune cells. Under the influence of hypoxia, the composition of these EVs is altered. In these EVs, increased levels of immunomodulatory proteins are frequently observed, resulting in immunosuppression and tumor persistence. Moreover, evidence reveals that modulation by miRNAs plays an important role in regulating the immune compartment of the tumor, where the delivery of modulators by hypoxic EVs mostly results in immunosuppression. In contrast, some EV-enriched factors may support antitumor immune responses. However, other regulatory mechanisms may be at play that prevent efficient uptake by the specific recipient immune cells, or the effects induced by these factors may not be sufficient to prevent tumor outgrowth.

## Figures and Tables

**Figure 1 cancers-14-04001-f001:**
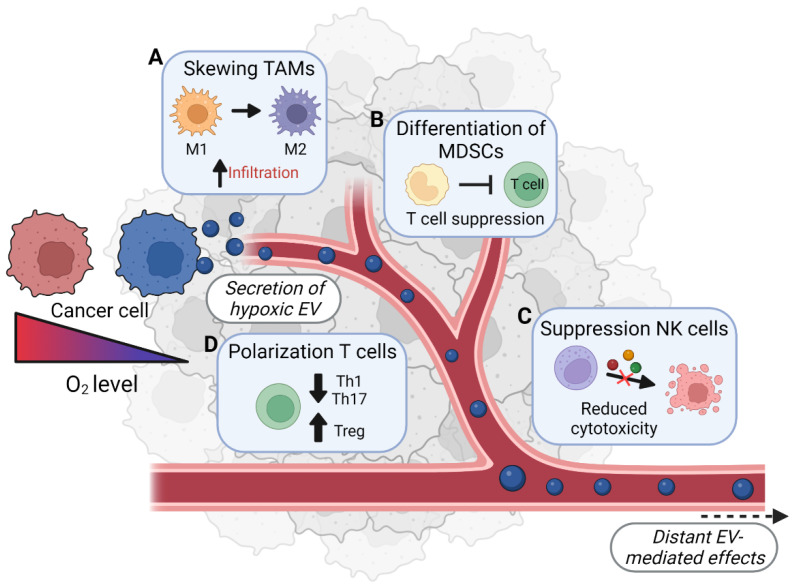
Suggested mechanisms for hypoxic tumor-derived EV-mediated immunosuppression to support tumor growth: Tumor hypoxia induces a more aggressive and therapy-resistant phenotype in cancer cells. This is associated with the release of immunosuppressive EVs, among other effects. These vesicles are taken up by a variety of immune cells, and prevent their differentiation into functional effector cells. In addition, these EVs induce the formation of regulatory immune cells such as Tregs and MDSCs, which further dampen the efficiency of the antitumor immune response. Furthermore, these EVs induce the formation of tumor-supportive M2 macrophages. Image created with BioRender.com, (accessed on 7 June 2022).

**Figure 2 cancers-14-04001-f002:**
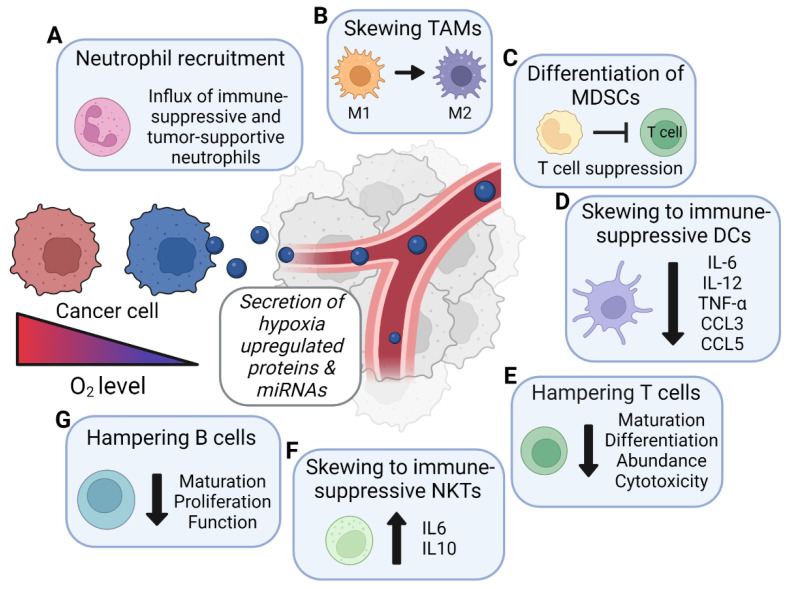
Hypoxia induces the secretion of immunosuppressive factors in EVs: Under hypoxia, cancer cells increase the secretion of various proteins and microRNAs via EVs. By themselves, these factors have been described to suppress immune reactions via inhibition of immune cells’ differentiation, proliferation, and effector function, and by inducing the differentiation of immunosuppressive cells. Secretion of these factors via EVs from hypoxic cancer cells might provide novel mechanisms by which these cells can manipulate their surroundings into providing a tumor-promoting environment. Image created with BioRender.com, (accessed on 7 June 2022).

**Table 1 cancers-14-04001-t001:** EV enriched factors and their effect on neutrophils.

Factor	Effect on Neutrophils	Proven to Be EV-Mediated? ^1^
CXCL8 (IL-8) [28,29]	Enhances recruitment to tumors through binding with CXCR1 and/or CXCR2. Reduces apoptosis and enhances proliferation via activation of MAPK signaling.	No
*MIR-451* [30]	Represses recruitment to tumors via inhibition of MAPK signaling.	No

^1^ Yes or no refers to the current knowledge regarding the respective hypoxic-EV-enriched proteins or miRNAs. Yes: a direct EV-mediated effect on the immune cell has been identified. No: no direct EV-mediated effects have yet been described, but the miRNA or protein has known immunomodulatory functions.

**Table 4 cancers-14-04001-t004:** EV-enriched factors and their effects on dendritic cells.

Factor	Effect on Dendritic Cells	Proven to Be EV-Mediated? ^1^
*MIR-301* [90]	Reduces production of the pro-inflammatory cytokines IL-6, IL-12, and TNF-α. Expression of miR-301 in DC represses the release of IFN-γ from DC-primed CD8^+^ and CD4^+^ responder cells.	No
*MIR-451* [91]	Reduces production of the pro-inflammatory cytokines IL-6, CCL3/MIP1a, CCL5/RANTES, and TNF-α.	No
CCL2 [92]	Reduces production of pro-inflammatory IL-12, hampering effective T-cell-mediated toxicity.	No

^1^ Yes or no refers to the current knowledge regarding the respective hypoxic EV-enriched proteins or miRNAs. Yes: a direct EV-mediated effect on the immune cell has been identified. No: no direct EV-mediated effects have yet been described, but the miRNA or protein has known immunomodulatory functions.

**Table 5 cancers-14-04001-t005:** EV-enriched factors and their effects on NK cells.

Factor	Effect on NK Cells	Proven to Be EV-Mediated? ^1^
TGF-β [97]	Inhibits NK cell function by decreasing surface expression of the activating receptor NKG2D and decreasing IFN-γ production.	Yes
*MIR-23a* [97]	Decreases the percentage of degranulating NK cells.	Yes

^1^ Yes or no refers to the current knowledge regarding the respective hypoxic EV-enriched proteins or miRNAs. Yes: a direct EV-mediated effect on the immune cell has been identified. No: no direct EV-mediated effects have yet been described, but the miRNA or protein has known immunomodulatory functions.

**Table 7 cancers-14-04001-t007:** EV-enriched factors and their effects on NKT cells.

Factor	Effect on NKT Cells	Proven to Be EV-Mediated? ^1^
*MIR-92a* [137]	Induces immunosuppressive NKT cells with reduced antitumor effects via increased IL-6 and IL-10 expression.	No

^1^ Yes or no refers to the current knowledge regarding the respective hypoxic EV-enriched proteins or miRNAs. Yes: a direct EV-mediated effect on the immune cell has been identified. No: no direct EV-mediated effects have yet been described, but the miRNA or protein has known immunomodulatory functions.

**Table 8 cancers-14-04001-t008:** EV-enriched factors and their effects on B cells.

Factor	Effect on B Cells	Proven to Be EV-Mediated? ^1^
*MIR-210* [139]	Impairs B-cell proliferation. Reduces antibody production.	No
*MIR-125* [73,140]	Prevents B cells’ maturation and release from the bone marrow.	No

^1^ Yes or no refers to the current knowledge regarding the respective hypoxic EV-enriched proteins or miRNAs. Yes: a direct EV-mediated effect on the immune cell has been identified. No: no direct EV-mediated effects have yet been described, but the miRNA or protein has known immunomodulatory functions.

**Table 9 cancers-14-04001-t009:** Immune stimulation by hypoxia-upregulated factors in EVs.

Factor	Immune-Stimulating Effect
*MIR-181*	Enhances the development of NK cells.
*MIR-155*	Enhances antitumor reactions in NK cells. Reduces the capacity of macrophages to respond to M2-inducing signals. Pushes macrophages towards the M1 phenotype. Modulates CD8^+^ T cells’ responsiveness to IL-2, IL-7, and IL-15 stimulation.

**Table 10 cancers-14-04001-t010:** Upregulated miRNAs in EVs under hypoxia.

miRNA	Cancer Type	Reported Culturing Conditions	Reported EV Isolation Method
*MIR-10* *MIR-21* *MIR-125* [73]	Glioma	1% O_2_, 48 h 10% EV-depleted FCS	Differential centrifugation (300× *g* 10 min, 2.000× *g* 10 min, 10,000× *g* 30 min) 0.22 μm filtration, ultracentrifugation (2 × 100,000× *g* 70 min).
*MIR-21-3p* *MIR-125b-5p* *MIR-181d-5p* [48]	Epithelial ovarian	1% O_2_, 24 h 10% EV-depleted FCS (100,000 g, 20 h)	Differential centrifugation (1000× *g* 10 min, 3000× *g* 30 min, Total Exosome Isolation Reagent (Life Technologies).
*MIR-23a* [112]	Lung	1% O_2_, 24 h 1% EV-free serum (Life Technologies)	Total Exosome Isolation Reagent (from cells) (Life Technologies).
*MIR-24-3p* [103]	Nasopharyngeal carcinoma	0.1% O_2_, 48 h	Differential centrifugation (300× *g* 10 min, 1.200× *g* 20 min, 10,000× *g* 30 min, 4 °C), 0.22 μm filtration, ultracentrifugation (2 × 100,000× *g* 3 h).
*MIR-92a* *MIR-127* *MIR-143* *MIR-181* *MIR-204* *MIR-292* *MIR-335* *MIR-433* *MIR-451* *MIR-542* *MIR-547a* *MIR-885* [126]	Prostate	1% O_2_, 72 h	Differential centrifugation at low speed (unspecified), ultracentrifugation at 30,000 RPM (type 70.1Ti fixed-angle rotor, L-80 Ultracentrifuge, Beckman Coulter).
*MIR-135b* [165]	Multiple myeloma	1% O_2_, 24 h serum-free medium	Centrifugation (3000× *g* 15 min), 0.22 μm PVDF filtration, ExoQuick Exosome Precipitation Solution (System Biosciences, Mountain View, CA).
*MIR-155* [166]	Hepatocellular carcinoma	1% O_2_, 24–72 h—CoCl_2_ 100 µM, 48 h 10% EV-depleted FCS (120,000 g overnight, 0.22 µm filtration)	Centrifugation (3000× *g* 15 min), 0.22 μm PVDF filtration, ExoQuick Exosome Precipitation Solution (System Biosciences, Mountain View, CA).
*MIR-210* [61]	Leukemia	1% O_2_, 24 h serum-free medium	Centrifugation (3000× *g* 15 min), 0.22 μm PVDF filtration, ExoQuick Exosome Precipitation Solution (System Biosciences, Mountain View, CA).
*MIR301a* [46]	Pancreas	1% O_2_ 10% EV depleted FCS	Differential centrifugation (300× *g* 10 min, 2000× *g* 10 min, 10,000× *g* 30 min, ultracentrifugation (100,000× *g* 70 min) OR ExoQuick Exosome Precipitation Solution.
*MIR-940* [47]	Epithelial ovarian	1% O_2_, 72 h 10% EV-depleted FCS (100,000 g, 20 h)	Centrifugation (2.500 RPM 30 min), Total Exosome Isolation Reagent (Invitrogen).
*MIR-1246* [44]	Glioma	1% O_2_ 10% EV depleted FCS	Differential centrifugation (300× *g* 10 min, 2000× *g* 10 min, 10,000× *g* 30 min), 0.22 μm filtration, ultracentrifugation (2 × 100,000× *g* 70 min).
*MIR-1273f* [167]	Hepatocellular carcinoma	1% O_2_, 48 h 10% EV-free FCS	Differential centrifugation (300× *g* 10 min, 2000× *g* 10 min, 10,000× *g* 30 min), ultracentrifugation (100,000× *g* 70 min).
*MIR-1290* [168]	Melanoma	1% O_2_, 72 h serum-free medium	Differential centrifugation (400× *g* 10 min, 2000× *g* 30 min), ultracentrifugation (110,000× *g* 70 min), flotation on an OptiPrep cushion (100,000× *g* 70 min), PBS wash, ultracentrifugation (110,000× *g* 70 min).
*MIR-135a* *MIR-494* *MIR-513a* *MIR-575* *MIR-1233-1* *MIR-4463* *MIR-4497* *MIR-4498* *MIR-4530* *MIR-4721* *MIR-4728* *MIR-4741* *MIR-4763* *MIR-6087* *MIR-6132* [169]	Melanoma	1% O_2_ serum-free medium	Differential centrifugation (300× *g* 4 min, 10,000× *g* 30 min), ultracentrifugation (2 × 100,000× *g* 2.5 h).
*Let-7a* [42]	Melanoma	<0.5% O_2_, 24 h 5% chemically defined medium (protein-free serum replacement)	Centrifugation (1.200× *g* 30 min), 300 kDa MWCO centrifugation at 4000 g, centrifugation (12,000× *g* 30 min), flotation on sucrose cushion (5.5% sucrose, 100,000× *g* 15 h).

**Table 11 cancers-14-04001-t011:** Upregulated proteins in EVs under hypoxia.

Protein	Cancer Type	Reported Culturing Conditions	Reported EV Isolation Method
ADAMTS1 [113]	Glioblastoma	<0.5% O_2_, 8–24 h, serum-free DMEM medium	CM centrifugation at 300× *g* for 10 min, 10,000× *g* for 30 min, and twice at 100,000× *g* for 2 h.
CCL2 (MCP1) [42]	Melanoma	<0.5% O_2_, 24 h 5% chemically defined medium (protein-free serum replacement)	CM centrifugation at 1.200× *g* for 30 min, 300 kDa MWCO centrifugation at 4000 g, 12,000× *g* for 30 min, and 100,000× *g* on a 5.5% sucrose pad for 15 h.
CSF-1 Ferritin heavy chain Ferritin light chain [42]	Melanoma	<0.5% O_2_, 24 h 5% chemically defined medium (protein-free serum replacement)	CM centrifugation at 1200× *g* for 30 min, 300 kDa MWCO centrifugation at 4000 g, 12,000× *g* for 30 min, and 100,000× *g* on a 5.5% sucrose pad for 15 h.
IGFBP1 IGFBP3 CXCL8 (IL-8) [27]	Glioma	1% O_2_, 48 h DMEM supplemented with 1% BSA	CM centrifugation at 300× *g* for 5 min, 16,500× *g* for 30 min, and 100,000× *g* for 2 h, and 2x PBS wash at 100,000× *g* for 2 h.
LOX [113]	Glioblastoma	<0.5% O_2_, 8–24 h serum-free DMEM	CM centrifugation at 300× *g* for 10 min, 10,000× *g* for 30 min, and twice at 100,000× *g* for 2 h.
Macrophage migration inhibitory factor (MIF) [42]	Melanoma	<0.5% O_2_, 24 h 5% chemically defined medium (protein-free serum replacement)	CM centrifugation at 1.200× *g* for 30 min, 300 kDa MWCO centrifugation at 4000 g, 12,000× *g* for 30 min, and 100,000× *g* on a 5.5% sucrose pad for 15 h.
PRMT5 [168]	Melanoma	1% O_2_, 72 h serum-free medium	CM centrifugation at 400× *g* for 10 min, 2000× *g* for 30 min, 100,000× *g* for 70 min, and 100,000× *g* for 70 min on an OptiPrep cushion.
TF [170]	Glioblastoma	1% O_2_, 30 min–48 h serum-free medium supplemented with 1% BSA (wt/vol).	CM centrifugation at 300× *g* for 10 min, 16,500× *g* for 20 min, 100,000× *g* for 2 h, and PBS washed at 100,000× *g* for 70 min.
TGF-β [97,114,115]	Park: Melanoma Berchem: Lung Rong: Breast	Park: <0.5% O_2_, 24 h 5% chemically defined medium (protein-free serum replacement). Berchem: 0.1% O_2_, 48 h exosome-depleted FBS. Rong: 1% O_2_, 4 days serum-free medium	Park: CM centrifugation at 1.200× *g* for 30 min, 300 kDa MWCO centrifugation at 4000× *g*, 12,000× *g* for 30 min, and 100,000× *g* on a 5.5% sucrose pad for 15 h. Berchem: CM centrifugation at 400× *g* for 5 min, 2.500× *g* for 20 min, 4.500 for 20 min, and 10,000× *g* for 1 h. Rong: CM centrifugation at 500× *g* for 2 × 10 min, 2000× *g* for 20 min, 10,000× *g* for 30 min, and 100,000× *g* for 1 h.
TSP-1 VEGF [113]	Glioblastoma	<0.5% O_2_, 8–24 h serum-free DMEM	CM centrifugation at 300× *g* for 10 min, 10,000× *g* for 30 min, and twice at 100,000× *g* for 2 h.
CAIX [116]	Renal-cell carcinoma	1% O_2_ or 200 μM CoCl2 advanced DMEM or advanced RPMI	CM centrifugation at 2000× *g* for 10 min and 12,000× *g* for 30 min; 0.22 μm PVDF filtration, and 70 min at 110,000 g, followed by density gradient centrifugation. Second method: Isolation by immunocapture Dynabeads conjugated with murine monoclonal anti-CD9 antibody.
Wnt4 [171,172]	Colorectal	250 μM Cocl2, 48 h exosome-depleted FBS	CM centrifugation at 1000 g for 10 min and 3000× *g* for 30 min. Added to Total Exosome Isolation Kit overnight and centrifuged at 10,000× *g* for 1 h.
MTA1 [42]	Melanoma	<0.5% O_2_, 24 h 5% chemically defined medium (protein-free serum replacement)	CM centrifugation at 1.200× *g* for 30 min, 300 kDa MWCO centrifugation at 4000 g, 12,000× *g* for 30 min, and 100,000× *g* on a 5.5% sucrose pad for 15 h.

Legend: CM = conditioned media.

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
