# Peer review of "The Immunomodulatory Role of Hypoxic Tumor-Derived Extracellular Vesicles"

_cancers, 2022, doi:10.3390/cancers14164001_

Round 1
Reviewer 1 Report
The review deals with the actual theme; hypoxic and cancer-derived extracellular vesicles (EV) and their role in the immunosuppression of cancers/tumors.
For tumor/cancer cells is hypoxia a key condition for their growth. Hypoxia induces a more aggressive and often therapy-resistant phenotype in these cells, thus contributing to the immune suppressive feature of the tumor microenvironment. Tumors communicate through EV with immune cells.
Review fills the information gap about the role of EV and miRNAs in tumor-associated immune cells (neutrophils, macrophages, myeloid-derived suppressor cells, dendritic cells, NK cells, T cells, NKT cells, B cells).
General comments:
The review is divided into 12 parts (Introduction, Immune cells, Importance of EV isolation methodology and experimental setup, Conclusion). The main body of the review includes a detailed description of the influence of EV on the different immune cells (neutrophils, macrophages, myeloid-derived suppressor cells, dendritic cells, NK cells, T cells, NKT cells, and B cells).
Authors are citing abundantly, references are in the frame of the selected field, recent, and cited correctly.
The main impact of the review contributes to knowledge of the detailed role of EV in cancer cells and under hypoxic conditions in tumors.
Specific comments:
To my opinion, the incorporation of the small part (few sentences) about the origin and biogenesis of EV (hypoxic, cancer) will improve the review and make it more informative.
Overall, I recommend this review for publication.
Author Response
Response to the comments of Reviewers
We greatly appreciate the overall positive response of the reviewers to our review. We made a comprehensive effort to edit the review and we specifically addressed the reviewer comments in a point-by-point response, which are included in this letter. We therefore hope that you will find our revised review acceptable for publication.
Reviewer 1
- To my opinion, the incorporation of the small part (few sentences) about the origin and biogenesis of EV (hypoxic, cancer) will improve the review and make it more informative. Overall, I recommend this review for publication.
We would like to refer the reviewer to lines 66-69, where we briefly explained the different biogenesis pathways for EV. These pathways are active over all cell types, including cancer cells. In lines 87-94, we expanded on the effect of hypoxia on the biogenesis of EV.
Specifically we added:"Hypoxia also affects EV biogenesis through regulation of RAB GTPases, that are involved in EV secretion (23, 24). For example, hypoxia-induced STAT3 regulates RAB proteins, promotes EV release by ovarian cancer cells and results in a more aggressive cancer phenotype (24). Although the exact mechanisms that control EV biogenesis in response to hypoxia remain to be clarified, recent evidence indicates that it is distinct of regular EV-biogenesis pathways. This is illustrated by difference in size and the dependence on GABAA receptor associated protein like 1 (GABARAPL1) (21, 22). "
Reviewer 2 Report
The review “The Immunomodulatory role of hypoxic tumor-derived extracellular vesicles” by Beaumont et al. is a comprehensive summary of the current knowledge about the role of extracellular vesicles derived from hypoxic tumor cells and their function in immune modulation in the tumor microenvironment. Extracellular vesicles are becoming more and more important in our understanding of communication processes within in the TME and have substantial impact on immunosuppression. The review is well written and summarizes the current knowledge about hypoxic tumor cell derived EVs and highlights in addition hypoxic cell-derived EV proteins or miRNAs with potential function in immunosuppression.
11) Figure 1 and figure 2: Please label the blue and red cell as cancer cell. Furthermore, the EVs released by the hypoxic tumor cell are all the same size. As you refer in your manuscript to MV and exosomes it would be beneficial if the depicted vesicles have different sizes.
22) Line 115: change “…tumor-supportive as tumor-suppressive…” to “…tumor-supportive as well as tumor-suppressive…”
33) In line 303 the authors describe 2 major subsets of DCs. Literature agrees on 3 major subsets (pDCs, cDC1 and cDC2). Please change accordingly. In addition, explain the abbreviation cDC. Furthermore, to clarify the function of cDC2 better the review would benefit if the authors would use in line 305 “preferentially” instead of “mainly”.
44) In line 383 the authors talk about pro-inflammatory type 1 cytokines. It would be helpful for the reader if the authors could add examples of type 1 and type 2 cytokines.
55) In table 6, MIR-24-3p is a misspelling. Change “ncreases FOXP3+…” to “Increases FOXP3+…”
66) For table 10 and 11 it would be helpful for the readers to add a column of the EV-producing cell type.
77)In table 11 an explanation of abbreviation “CM” needs to be added.
Author Response
Response to the comments of Reviewers
We greatly appreciate the overall positive response of the reviewers to our review. We made a comprehensive effort to edit the review and we specifically addressed the reviewer comments in a point-by-point response, which are included in this letter. We therefore hope that you will find our revised review acceptable for publication.
Review 2
- Figure 1 and figure 2: Please label the blue and red cell as cancer cell. Furthermore, the EVs released by the hypoxic tumor cell are all the same size. As you refer in your manuscript to MV and exosomes it would be beneficial if the depicted vesicles have different sizes.
We added the labels for the cancer cells. We also changed the EV symbols to varying sizes, which captures the heterogenous sizes of EV.
- Line 115: change “…tumor-supportive as tumor-suppressive…” to “…tumor-supportive as well as tumor-suppressive…
This is changed as suggested. Highlighted adaptation on line 122.
- In line 303 the authors describe 2 major subsets of DCs. Literature agrees on 3 major subsets (pDCs, cDC1 and cDC2). Please change accordingly.
We included background on the pDCs. Highlighted adaptation on lines 312 and 317-322.
In addition, explain the abbreviation cDC.
We explained the abbreviation in the first sentence of the paragraph. Highlighted adaptation on line 308.
Furthermore, to clarify the function of cDC2 better the review would benefit if the authors would use in line 305 “preferentially” instead of “mainly”.
This is changed as suggested. Highlighted adaptation on line 315.
- In line 383 the authors talk about pro-inflammatory type 1 cytokines. It would be helpful for the reader if the authors could add examples of type 1 and type 2 cytokines.
As suggested, we added examples of the respective cytokine subtypes.Highlighted adaptations on lines 392-393 and 394-395.
- In table 6, MIR-24-3p is a misspelling. Change “ncreases FOXP3+…” to “Increases FOXP3+…”
This is changed as suggested. Highlighted adaptation in table 6.
- For table 10 and 11 it would be helpful for the readers to add a column of the EV-producing cell type.
We agree with this suggestion and we added a column titled: ‘Cancer type’.
- In table 11 an explanation of abbreviation “CM” needs to be added.
This is changed as suggested. Highlighted adaptation in Legend table 11.